# Learning from multiscale wavelet superpixels using GNN with spatially heterogeneous pooling

## Abstract

Neural networks have become the standard for image classification tasks. On one hand, convolutional neural networks (CNNs) achieve state-of-the-art performance by learning from a regular grid representation of images. On the other hand, graph neural networks (GNNs) have shown promise in learning image classification from an embedded superpixel graph. However, in the latter, studies have been restricted to SLIC superpixels, where 1) a single target number of superpixels is arbitrarily defined for an entire dataset irrespective of differences across images and 2) the superpixels in a given image are of similar size despite intrinsic multiscale structure. In this study, we investigate learning from a new principled representation in which individual images are represented by an image-specific number of multiscale superpixels. We propose WaveMesh, a wavelet-based superpixeling algorithm, where the number and sizes of superpixels in an image are systematically computed based on the image content. We also present WavePool, a spatially heterogeneous pooling scheme tailored to WaveMesh superpixels. We study the feasibility of learning from the WaveMesh superpixel representation using SplineCNN, a state-of-the-art network for image graph classification. We show that under the same network architecture and training settings, SplineCNN with original Graclus-based pooling learns from WaveMesh superpixels on-par with SLIC superpixels. Additionally, we observe that the best performance is achieved when replacing Graclus-based pooling with WavePool while using WaveMesh superpixels.

## 1 Introduction

Convolutional neural networks (CNNs) achieve state-of-the-art performance on a variety of image classification tasks from different domains (Tan & Le, 2019; Gulshan et al., 2016). CNNs learn from a regular pixel-grid representation of the images. Although not all pixels provide equal amount of new information, by design the filters in the first layer of a CNN operate on each pixel from top-left to bottom-right in the same way. Additionally, images are typically resized to a prescribed size before feeding into a CNN. In applications that use standard CNN architectures or pre-trained models on a new image classification dataset, the images are typically uniformly downsampled to meet the input size requirements of the architecture being used. Uniform downsampling may be suboptimal as real data naturally exhibits spatial and multiscale heterogeneity. Few studies have explored the impact of input image resolution on model performance (Sabottke & Spieler, 2020), despite its recognized importance (Lakhani, 2020).

Graph neural network (GNN) is a type of neural network that learns from graph structured data. Recent studies have shown the performance of GNNs on image graph classification tasks (Monti et al., 2017; Fey et al., 2018; Knyazev et al., 2019; Dwivedi et al., 2020). In this task, a GNN learns to classify images from embedded graphs that represent superpixels in the images. However, prior studies have been restricted to SLIC superpixels (Achanta et al., 2012). In this framework, a single target number of superpixels is arbitrarily defined for an entire dataset irrespective of differences across images, and the superpixels in a given image are of similar size despite intrinsic multiscale structure. Our proposed approach circumvents these limitations, as shown in Figure 1.

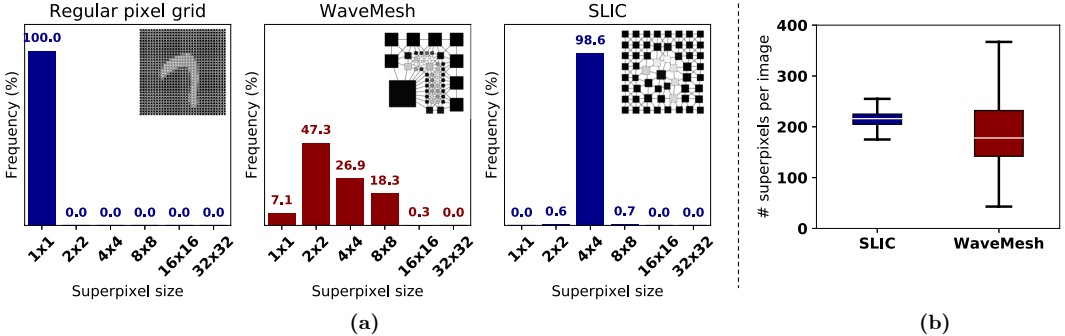

Figure 1: (a) Average distribution of superpixel size averaged across MNIST training dataset for different superpixel representation: none (left), WaveMesh (center), and SLIC (right). In each panel, an insert shows the graph representation of a single sample for illustration. Size of a node in the graph is proportional to the superpixel size. SLIC superpixels are not cubic yet the x-axis binning is chosen to match other plots. (b) Boxplots of the # superpixels per image for CIFAR-10 training dataset.

The objectives of our work are twofold. First, we aim to rethink the process of downsampling and/or superpixeling images by introducing a multiscale superpixel representation that can be considered as in between the regular grid and similar-sized superpixel representations. Secondly, we systematically study the feasibility of learning to classify images from embedded graphs that represent the multiscale superpixels. In this context, the contributions of our study are as follows.

- We present WaveMesh, an algorithm to superpixel (compress) images in the pixel domain. WaveMesh is based on the quadtree representation of the wavelet transform. Our sample-specific method leads to non-uniformly distributed and multiscale superpixels. The number and size of superpixels in an image are systematically computed by the algorithm based on the image content. WaveMesh requires at most one tunable parameter.
- We propose WavePool, a spatially heterogeneous pooling method tailored to WaveMesh superpixels. WavePool preserves spatial structure leading to interpretable intermediate outputs. WavePool generalizes the classical pooling employed in CNNs, and easily integrates with existing GNNs.
- To evaluate the WaveMesh representation and the WavePool method for image graph classification, we compare them with SLIC superpixels and graclus-based pooling by conducting several experiments using SplineCNN, a network proposed by Fey et al. (2018).

## 2   RELATED WORK

**Superpixeling.** Grouping pixels to form superpixels was proposed by Ren & Malik (2003) as a preprocessing mechanism that preserves most of the structure necessary for image segmentation. Since then many superpixeling algorithms have been proposed including deep learning based methods (Liu et al., 2011; Li & Chen, 2015; Tu et al., 2018; Giraud et al., 2018; Yang et al., 2020; Zhang et al., 2020). The SLIC algorithm proposed by Achanta et al. (2012) is based on $k$-means clustering.

**GNN for image graph classification.** Prior studies have demonstrated the representational power and generalization ability of GNNs on image graph classification tasks using SLIC superpixels. Dwivedi et al. (2020) show that message passing graph convolution networks (GCNs) outperform Weisfeiler-Lehman GNNs on MNIST and CIFAR-10 datasets. Recognizing the importance of spatial and hierarchical structure inherent in images, Knyazev et al. (2019) model images as multigraphs that represent SLIC superpixels computed at different user-defined scales, and then successfully train GNNs on the multigraphs. SplineCNN proposed by Fey et al. (2018) is another network for learning from irregularly structured data. It builds on the work of Monti et al. (2017), but uses a spline convolution kernel instead of Gaussian mixture model kernels.

**Graclus-based pooling.** Pooling is used in GNNs to coarsen the graph by aggregating nodes within specified clusters. Graclus is a kernel-based multilevel graph clustering algorithm that efficiently clusters nodes in large graphs without any eigenvector computation. Graclus is used in many GNNs to obtain a clustering on the nodes, which is then used by the pooling operator to coarsen the graph

(Defferrard et al., 2016; Monti et al., 2017; Fey et al., 2018). Hereafter, we refer to pooling based on graclus clustering as graclus-based pooling.

# 3 WaveMesh: Multiscale Wavelet Superpixels

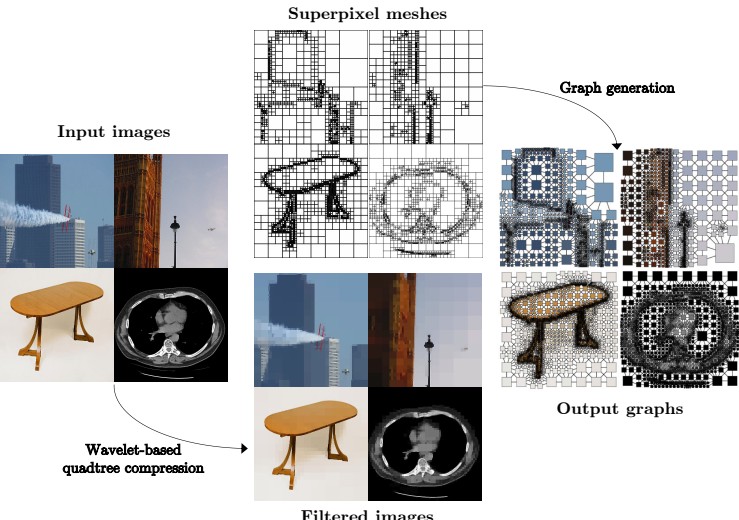

Figure 2: Filtering images in wavelet space generates non-uniform superpixel meshes that are then represented as embedded graphs. The leftmost images are preprocessed with the method described in section 3 with a threshold value equal to five times the theoretical value. Natural images are from the Pascal dataset (Everingham et al., 2010), and the medical image is from the NLST dataset.

The WaveMesh algorithm is broken down into its elementary steps below: 1) images are wavelet transformed, 2) images are filtered in wavelet space by thresholding the wavelet coefficients, and 3) the superpixel mesh is generated from the wavelet-filter mask. The algorithm is rooted in wavelet theory's seminal work (Mallat, 1989; Donoho & Johnstone, 1994b). The particular way in which wavelets are used in this work is inspired by their related application in the physical sciences (Schneider & Vasilyev, 2010; Bassenne et al., 2017; 2018).

## 3.1 Step 1: Wavelet transformation of the input image

Consider a two-dimensional (2D) image $I$ discretely described by its pixel values $I[\boldsymbol{x_0}]$ centered at locations $\boldsymbol{x_0} = 2^{-1}(i\Delta, j\Delta)$ of a $N \times N$ regular grid, where $\Delta$ is the inter-pixel spacing and $(i,j) = 1, 3, \ldots, 2N - 1$. A continuous wavelet representation of $I$ is $I(\boldsymbol{x}) = \sum_{\boldsymbol{x_0}} \widehat{I}^{(0)}[\boldsymbol{x_0}]\phi^{(0)}(\boldsymbol{x} - \boldsymbol{x_0})$, where $\boldsymbol{x}$ is the continuous pixel-space coordinate, and $\phi^0(\boldsymbol{x} - \boldsymbol{x_0})$ are scaling functions that form a orthonormal basis of low-pass filters centered at $\boldsymbol{x_0}$, with filter width $\Delta$. The scaling functions have unit energy $\langle \phi^0(\boldsymbol{x} - \boldsymbol{x_0})\phi^0(\boldsymbol{x} - \boldsymbol{x_0}) \rangle = 1$, where the bracket operator $\langle y \rangle = 1/(N\Delta)^2 \int y(\boldsymbol{x})d\boldsymbol{x}$ denotes the global average for a general 2D continuous field $y(\boldsymbol{x})$. In practice, when dealing with discrete signals, $\widehat{I}^{(0)}[\boldsymbol{x_0}]$ cannot be computed exactly, since $I$ is only known at discrete points $\boldsymbol{x_0}$. Instead, it is numerically discretized and the approximation coefficients $\widehat{I}^{(0)}[\boldsymbol{x_0}]$ are estimated as an algebraic function of $I[\boldsymbol{x_0}]$. Assuming that $\phi^0(\boldsymbol{x} - \boldsymbol{x_0})$ decays fast away from $\boldsymbol{x} = \boldsymbol{x_0}$, we get $\widehat{I}^{(0)}[\boldsymbol{x_0}] = I[\boldsymbol{x_0}]/N$ (Addison, 2017). This estimate for $\widehat{I}^{(0)}[\boldsymbol{x_0}]$ is the initialization stage of the recursive wavelet multiresolution algorithm (MRA) of Mallat (1989), which enables the computation of wavelet coefficients at coarser scales.

The decomposition of the finest-scale low-pass filter $\phi^0(\boldsymbol{x} - \boldsymbol{x_0})$ in terms of narrow-band wavelet filters $\psi^{(s,d)}(\boldsymbol{x} - \boldsymbol{x_s})$ with increasingly large filter width and a coarsest-scale scaling function

$\phi^{(S)}(\boldsymbol{x} - \boldsymbol{x_S})$ yields the full wavelet-series expansion of $I$,

$$I(\boldsymbol{x}) = \sum_{s=1}^{S}\sum_{\boldsymbol{x_s}}\sum_{d=1}^{3} \breve{I}^{(s,d)}[\boldsymbol{x_s}]\psi^{(s,d)}(\boldsymbol{x}-\boldsymbol{x_s}) + \widehat{I}^{(S)}[\boldsymbol{x_S}]\phi^{(S)}(\boldsymbol{x}-\boldsymbol{x_S}). \tag{1}$$

Here, $\breve{I}^{(s,d)}[\boldsymbol{x_s}] = \left\langle I(\boldsymbol{x})\psi^{(s,d)}(\boldsymbol{x}-\boldsymbol{x_s})\right\rangle$ and $\widehat{I}^{(S)}[\boldsymbol{x_S}] = \left\langle I(\boldsymbol{x})\phi^{(S)}(\boldsymbol{x}-\boldsymbol{x_S})\right\rangle$ are wavelet and approximation coefficients at scale $s$ and $S$, respectively, obtained from the orthonormality properties of the wavelet and scaling functions. In this formulation, $d = (1, 2, 3)$ is a wavelet directionality index, and $s = (1, 2 \ldots, S)$ is a scale exponent, with $S = \log_2 N$ the number of resolution levels allowed by the grid (5 for 32×32 images). Similarly, $\boldsymbol{x_s} = 2^{s-1}(i\Delta, j\Delta)$ is a scale-dependent wavelet grid of $(N/2^s)\times(N/2^s)$ elements where the basis functions are centered, with $i, j = 1, 3, \ldots, N/2^{s-1} - 1$. The wavelet coefficients represent the local fluctuations of $I$ centered at $\boldsymbol{x_s}$ at scale $s$, while the approximation coefficient is proportional to the global mean of $I$. At each scale, the filter width of the wavelets is $2^s\Delta$.

In this study, the 2D orthonormal basis functions $\psi^{(s,d)}(\boldsymbol{x} - \boldsymbol{x_s})$ are products of one-dimensional (1D) Haar wavelets (Meneveau, 1991). The definition of 2D wavelets as multiplicative products of 1D wavelets is a particular choice that follows the MRA formulation (Mallat, 1989). Haar wavelets have a narrow spatial support that provides a high degree of spatial localization. However, they display large spectral leakage at high wavenumbers since infinite spectral and spatial resolutions cannot be simultaneously attained due to limitations imposed by the uncertainty principle (Addison, 2017). Different boundary conditions can be assumed for the field $I$. We do not require such a choice in this study as we restrict ourselves to square images. However, the wavelet MRA framework is not limited to square inputs and can be generalized to rectangular inputs (Addison, 2017; Kim et al., 2018).

The definition of 2D wavelets as multiplicative products of 1D wavelets is a particular choice that follows the MRA formulation described by Mallat (1989), in which, the multivariate wavelets are characterized by an isotropic scale and therefore render limited information about anisotropy in the image. A large number of alternative basis functions have been recently proposed for replacing traditional wavelets when analyzing multi-dimensional data that exhibit complex anisotropic structures such as filaments and sheets. These include, but are not limited to, curvelets, contourlets, and shearlets (see Kutyniok & Labate (2012) for an extensive review on this topic).

### 3.2 STEP 2: FILTERING OF THE IMAGE IN WAVELET SPACE

The second step decomposes $I$ as

$$I = I_> + I_\le, \tag{2}$$

where the filtered $I_>$ and remainder $I_\le$ components correspond to the highest and lowest energetic wavelet modes of $I$, respectively. By construction, these two components are not spatially cross-correlated, as implied by the orthogonality of the wavelets and by the filtering operation described below. Note that large wavelet coefficients are associated with large fluctuations within the corresponding region of the scale-dependent wavelet grid $\boldsymbol{x_s}$, these being markers of underlying coherent structures. Under the assumptions that $I_\le$ is additive Gaussian white noise, Donoho & Johnstone (1994a) described a wavelet-based algorithm that is optimal for achieving the target decomposition (2), since it minimizes the maximum $\mathbb{L}^2$-estimation error of $I_>$. $I_>$ is obtained by retaining only the wavelet coefficients $\breve{I}^{(s,d)}$ whose absolute values satisfy

$$\breve{I}_>^{(s,d)}(\boldsymbol{x_s}) = \begin{cases} \breve{I}^{(s,d)}(\boldsymbol{x_s}) & \text{if } |\breve{I}^{(s,d)}(\boldsymbol{x_s})| \ge T, \\ 0 & \text{otherwise}, \end{cases} \tag{3}$$

for all scales $s$, positions $\boldsymbol{x_s}$ and directions $d$. In Equation 3, $T$ is a theoretical threshold defined as

$$T = \sqrt{2\sigma_{I_\le}^2 \ln N^2}, \tag{4}$$

where $\sigma_{I_\le}^2$ is the unknown variance of $I_\le$. In this study, the iterative method of Azzalini et al. (2005) is employed, which converges to $T$ starting from a first iteration where $\sigma_{I_\le}^2$ in Equation 4 is substituted by the variance $\sigma_I^2$ of the total image $I$. This iterative procedure does not introduce significant computational overhead, since only one wavelet transform is required independently of the number of iterations. The algorithm does not introduce any hyperparameter when the theoretical threshold value is used. Note that the threshold is image-dependent, thereby ensuring that the algorithm adapts

the number of superpixels to each image appropriately. The above filtering operation is equivalent to applying a binary filter mask to wavelet coefficients, denoted as wavelet-filter mask below.

The iterative method is deemed as converged when the relative variation in the estimated threshold $T$ is less than $0.1\%$ across consecutive iterations. A maximum of $\mathcal{O}(10)$ iterations were required to obtain the results presented below. The overall computational cost is $\mathcal{O}(n_i M)$, where $n_i$ is the number of iterations and $M$ is the number of pixels in the image (Azzalini et al., 2005). In this work, we allow for further reduction in number of superpixels by varying the threshold $T$ to take larger values. Figure 2 illustrates the application of this wavelet filtering method on four images, wherein for RGB images filtering is applied to each channel independently. Most of the structural and edge information is preserved at all scales. However, a drawback of the method is that the superpixel boundaries are necessarily regular and axis-aligned.

### 3.3 STEP 3: GENERATING THE SUPERPIXEL MESH FROM THE WAVELET-FILTER MASK

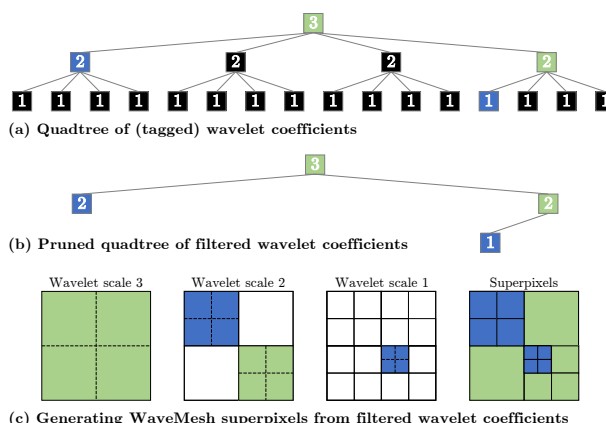

(a) Quadtree of (tagged) wavelet coefficients

(b) Pruned quadtree of filtered wavelet coefficients

(c) Generating WaveMesh superpixels from filtered wavelet coefficients

Figure 3: Illustration of the wavelet-based quadtree compression algorithm for an $8{\times}8$ image, along with the resulting adapted grid. Starting from the coarsest possible wavelet grid that contains just one superpixel, the algorithm adapts the grid by recursively splitting it. If the wavelet coefficient corresponding to a region is tagged (denoted by blue color), then that region is split into $2{\times}2$ superpixels.

To generate superpixels for a given image, the final step is a grid adaptation based on the wavelet-filter mask described in subsection 3.2. The result is a non-uniform grid of multiscale superpixels adapted around regions of the image with high variability.

**Quadtree representation.** The algorithm is perhaps best understood by representing the wavelet coefficients in a quadtree (Finkel & Bentley, 1974), a tree data structure in which each node has exactly four children. A quadtree-based representation of wavelet coefficients was previously shown to be an efficient data structure for wavelet-based image compression (Banham & Sullivan, 1992; Wakin et al., 2003). Here, the height of this quadtree equals the number of decomposition levels $S$ in the wavelet transform. Each vertex at a given level $s$ is associated with a triplet of wavelet coefficients $[\breve{I}^{(s,d=0)}(\boldsymbol{x_s}), \breve{I}^{(s,d=1)}(\boldsymbol{x_s}), \breve{I}^{(s,d=2)}(\boldsymbol{x_s})]$. All vertices from a given level correspond to wavelet coefficients across all locations at a given scale. The children vertices of a root vertex are the wavelet coefficients from that region in space at smaller scales. The quadtree representation of the wavelet coefficients of an $8{\times}8$ image is schematically represented in Figure 3(a). The number on each vertex indicates the scale, from the smallest scale $s{=}1$ associated with $2{\times}2$ pixel patches up to the largest scale $s{=}3$ associated with the entire $8{\times}8$ image. The pixel regions associated with each wavelet coefficient are delineated by solid lines in the three leftmost figures in Figure 3(c).

**Node tagging.** The vertices in the tree are tagged according to the filtering algorithm described in subsection 3.2. The tagged elements of the tree denoted by blue filled color in Figure 3(a,b) correspond to those with absolute values larger than the threshold $T$, and therefore correspond to locations in the image with important spatial variability. In the 2D case, tagging is applied if at least one of the 3 wavelet coefficients of $I$ per location is larger than the threshold. Additional tagging

by green-filled color is applied to wavelet coefficients that are smaller than the threshold $T$ but that correspond to a spatial region with at least one tagged wavelet coefficient at a smaller scale. This corresponds to tagging all the ancestors of previously tagged vertices. This tagging procedure enforces cubic superpixels by ensuring that when there is a coherent structure at scale $s$ but not at a larger scale $s+1$, the wavelet coefficient at scale $s+1$ at that location are also tagged, hence triggering local grid refinement at level $s+1$. Non-tagged vertices are pruned as shown in Figure 3(b).

**Mesh generation.** Starting from the coarsest possible wavelet grid $x_s = x_S$ that contains just one superpixel, the algorithm adapts the grid by recursively splitting it as follows. If the wavelet coefficient corresponding to a region is tagged, then that region is split into $2\times2$ superpixels, which locally refines the grid. The algorithm is stopped otherwise. The same recursive loop is then applied to the refined superpixels. The final configuration of the adapted grid is obtained when none of the wavelet coefficients in any the superpixels are tagged. An example of final adapted grid is shown in Figure 3(c). The dashed lines correspond to the superpixel refinement due to the vertex being tagged. Adapted grids from real images are shown in Figure 2 where the superpixel meshes exhibit desired level of heterogeneity with multiscale refinement around edges. For RGB images, the most restrictive mesh is employed at every location and scale. In other words, tagging for the full image is applied if at least of the channels is tagged.

## 4 WavePool: Spatially heterogeneous pooling

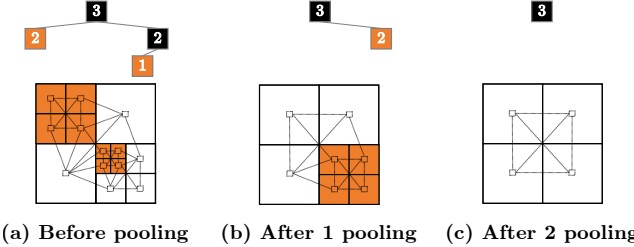

(a) Before pooling    (b) After 1 pooling    (c) After 2 pooling

Figure 4: Illustration of WavePool from wavelet quadtree representation. Leaf nodes ($2\times2$ superpixels) are recursively pooled. In the lower panel, dashed squares and lines correspond to nodes and edges in the region adjacency graph (RAG) representation of the superpixel mesh, respectively.

The proposed spatially heterogeneous pooling, WavePool, is best explained using the wavelet coefficient quadtree representation described in subsection 3.3. One WavePool operation consists in aggregating all the leaf nodes of the wavelet quadtree. In the pixel domain, this step essentially corresponds to merging patches of $2\times2$ superpixels into a parent superpixel, and aggregating the node features with a choice of pooling function (mean or max typically). Figure 4 illustrates WavePool on a simple superpixel mesh and its effect on both the quadtree (Figure 4 upper panel) and region adjacency graph (Figure 4 lower panel) representation. In a region adjacency graph (RAG), nodes correspond to superpixels, and edges connect neighboring superpixels. We show RAG in Figure 4 because we train GNNs to learn from embedded RAGs. RAG is not a tree and should not be confused with the wavelet coefficient quadtree.

By construction, WavePool generalizes the classical CNN pooling operation. For a regular superpixel grid as in Figure 5, WavePool exactly matches the conventional $2\times2$ pooling in CNN. Although more general than its CNN counterpart, WavePool is restricted to WaveMesh superpixels or more broadly to any quadtree based superpixel representation (Tanimoto & Pavlidis (1975); Zhang et al. (2018)), unlike graclus-based pooling. However, graclus-based pooling does not converge to CNN pooling even when applied to regular superpixel grids as shown in Figure 5.

## 5 Experiments and Results

**Datasets.** We performed image graph classification experiments on SLIC and WaveMesh superpixels from 3 datasets: MNIST, Fashion-MNIST, and CIFAR-10 (LeCun et al., 1998; Xiao et al., 2017; Krizhevsky et al., 2009). We represent superpixels by embedded region adjacency graphs (RAG),

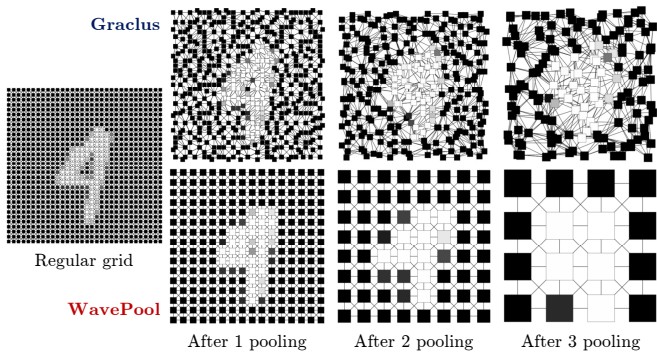

Figure 5: Illustration of WavePool versus graclus-based pooling on a regular superpixel grid.

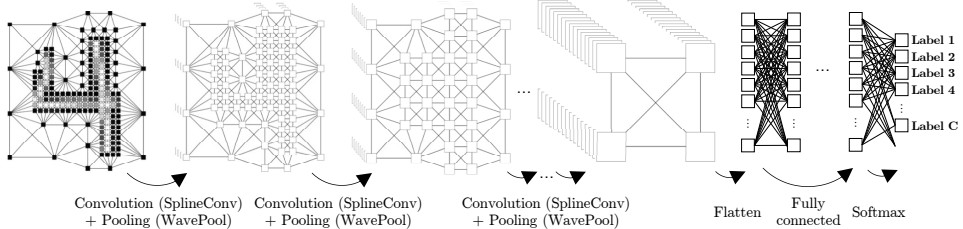

Figure 6: Model architecture. WaveMesh superpixel graph of an example image (digit 4) from MNIST dataset passing through SplineCNN with WavePool. Nodes represent superpixel centers.

where nodes correspond to superpixels, and edges connect neighboring superpixels. Node embeddings are mean intensity of superpixels. Edges in the graph are directed with pseudo-coordinates as in Fey et al. (2018). For more details on the datasets, refer to subsection A.1.

**EXPERIMENTAL SETTINGS.** We conduct experiments on two configurations based on a SplineCNN implementation available in PyTorch Geometric (Fey & Lenssen, 2019). The configurations are:

1. SplineConv($(3, 3), 1, 32$) –> Pool –> SplineConv($(3, 3), 32, 64$) –> Pool –> Global mean pool –> FC($128$) –> FC($10$). This configuration has **30506 parameters**.
2. SplineConv($(3, 3), 1, 32$) –> Pool –> SplineConv($(3, 3), 32, 64$) –> Pool –> SplineConv($(3, 3), 64, 128$) –> Pool –> Global mean pool –> FC($256$) –> FC($10$). This configuration has **139178 parameters**.

Through the experiments we aim to: 1) Compare how SplineCNN performs on SLIC and WaveMesh superpixels under the same network architecture and training settings; 2) Understand the effect of WavePool on learning from WaveMesh superpixels, everything else being the same. The PyTorch Geometric implementation of SplineCNN uses Adam optimizer with an initial learning rate of 0.01, which is decreased by a factor of 10 after 15 and 25 epochs. Since the goal of our experiments is not tune the best model for WaveMesh superpixel representation, we conducted all experiments without any hyperparameter tuning. We use the same training settings and train the network for 30 epochs on MNIST and Fashion-MNIST, and for 75 epochs on CIFAR-10. The pooling function is max for both WavePool and graclus-based pooling. All experiments are repeated 5 times, and the mean train and test accuracy are reported along with the standard deviation. Figure 6 is a visual illustration of a WaveMesh superpixel graph passing through SplineCNN network with WavePool.

**MNIST.** Results on the MNIST dataset from our experiments and prior work are shown in Table 1. We report the mean and standard deviation values for train and test accuracy, and precision. We didn't include recall since averaged one-versus-all recall and accuracy are equal for balanced datasets. Experiments 1–4 uses WaveMesh superpixels obtained using the theoretical threshold $T$ as described in subsection 3.2. Experiments 3 and 4 are same as 1 and 2, but uses a network (config 2) with more parameters. Across these four experiments we first observe that SplineCNN is successful in learning to classify images from the WaveMesh representation. We also observe that the network with WavePool performs better than the one with graclus-based pooling. Experiments 7–10 are

same as 1–4 but with lesser WaveMesh superpixels. These experiments were done to compare with experiments 13–14 that report results from prior work on SLIC superpixels where each image has exactly 75 superpixels. To reduce the number of WaveMesh superpixels in an image to about 75, we increased the theoretical threshold $T$ in our algorithm. From the results for experiments 7–10, we observe that the network with WavePool performs similar to or better than graclus-based pooling. Experiments 5–6 and 11–12 are on SLIC superpixels that we generated using the scikit-learn package. Comparing experiments 1–4 with 5 and 6, and 7–10 with 11 and 12, we observe that SplineCNN learns just as well or better from WaveMesh superpixels.

| # | Superpixel | #Nodes | Config | Pooling | Train acc | Test acc | Precision |
|---|-----------|--------|--------|---------|-----------|----------|-----------|
| 1 | WaveMesh | 238±50 | 1 | Graclus | 92.33±0.09 | 89.63±0.45 | 89.71±0.44 |
| 2 | WaveMesh | 238±50 | 1 | WavePool | 95.75±0.08 | **95.44±0.12** | **95.40±0.13** |
| 3 | WaveMesh | 238±50 | 2 | Graclus | 98.39± 0.05 | 96.80±0.11 | 96.86±0.08 |
| 4 | WaveMesh | 238±50 | 2 | WavePool | 99.68± 0.03 | **98.68±0.08** | **98.67±0.08** |
| 5 | SLIC | 241±5 | 1 | Graclus | 95.50±0.21 | 95.51±0.29 | 95.48±0.30 |
| 6 | SLIC | 241±5 | 2 | Graclus | 98.07±0.04 | 97.83±0.11 | 97.83±0.11 |
| 7 | WaveMesh | 57±12 | 1 | Graclus | 93.34±0.04 | 92.53±0.15 | 92.47±0.15 |
| 8 | WaveMesh | 57±12 | 1 | WavePool | 96.30±0.10 | **93.74±0.17** | **93.72±0.16** |
| 9 | WaveMesh | 57±12 | 2 | Graclus | 95.68±0.09 | **94.21±0.21** | **94.24±0.22** |
| 10 | WaveMesh | 57±12 | 2 | WavePool | 99.23±0.04 | 93.84±0.48 | 93.93±0.40 |
| 11 | SLIC | 59±2 | 1 | Graclus | 92.34±0.11 | 91.18±0.22 | 91.13±0.21 |
| 12 | SLIC | 59±2 | 2 | Graclus | 94.13±0.08 | **92.99±0.22** | **93.01±0.20** |
| 13 | SLIC (Monti et al., 2017) | 75±0 | – | Graclus | – | 91.11 | – |
| 14 | SLIC (Fey et al., 2018) | 75±0 | – | Graclus | – | 95.22 | – |

Table 1: Results on MNIST dataset. Mean±SD (%) are reported for each evaluation metric.

| # | Superpixel | #Nodes | Config | Pooling | Train acc | Test acc | Precision |
|---|-----------|--------|--------|---------|-----------|----------|-----------|
| 1 | WaveMesh | 436±129 | 1 | Graclus | 80.36±0.39 | 65.35±2.94 | **73.84±2.87** |
| 2 | WaveMesh | 436±129 | 1 | WavePool | 85.77±0.18 | **76.60±0.83** | **79.34±0.49** |
| 3 | WaveMesh | 436±129 | 2 | Graclus | 85.40±0.10 | 75.69±1.47 | 79.40±0.36 |
| 4 | WaveMesh | 436±129 | 2 | WavePool | 92.58±0.07 | **83.66±1.49** | **83.86±1.53** |
| 5 | WaveMesh | 261±35 | 1 | Graclus | 81.32±0.13 | 76.75 ± 0.33 | 77.78±0.33 |
| 6 | WaveMesh | 261±35 | 1 | WavePool | 85.91±0.10 | **81.35±0.69** | **81.99±0.48** |
| 7 | WaveMesh | 261±35 | 2 | Graclus | 85.18±0.18 | 79.78±0.46 | 80.47±0.52 |
| 8 | WaveMesh | 261±35 | 2 | WavePool | 92.34±0.15 | **87.65±0.36** | **87.66±0.32** |
| 9 | SLIC | 259±7 | 1 | Graclus | 82.91±0.08 | 81.49±0.38 | 81.21±0.41 |
| 10 | SLIC | 259±7 | 2 | Graclus | 86.71±0.10 | **85.00±0.32** | **84.86±0.35** |
| 11 | WaveMesh | 134±22 | 1 | Graclus | 80.92±0.16 | 78.85±0.09 | 78.87±0.06 |
| 12 | WaveMesh | 134±22 | 1 | WavePool | 85.18±0.13 | **80.42±0.33** | **80.80±0.29** |
| 13 | WaveMesh | 134±22 | 2 | Graclus | 83.84±0.13 | 80.60±0.21 | 80.97±0.41 |
| 14 | WaveMesh | 134±22 | 2 | WavePool | 90.98±0.13 | **82.65±0.52** | **82.95±0.64** |
| 15 | SLIC | 118±4 | 1 | Graclus | 81.46±0.16 | 79.59±0.34 | 79.18±0.34 |
| 16 | SLIC | 118±4 | 2 | Graclus | 84.11±0.15 | **82.12±0.27** | **82.08±0.23** |
| 17 | SLIC (Avelar et al., 2020) | ≤ 75 | – | – | – | 83.07 | – |

Table 2: Results on Fashion-MNIST. Mean±SD (%) are reported for each evaluation metric.

**FASHION-MNIST AND CIFAR-10.** Experiments similar to MNIST were performed on these two datasets. Results are reported in Tables 2 and 3. For both datasets, experiments 1–4 were performed on WaveMesh superpixels obtained using the theoretical threshold $T$ in our algorithm. In Table 3, experiments 7–9 report results from (Dwivedi et al., 2020), where RingGNN and Gated GCN perform the worst and best. Results for MoNet are shown because SplineCNN builds on the work of MoNet. For the case of Fashion-MNIST, experiments 1–4 with more superpixels have similar train accuracy as in experiments 5–8. However, the trained model in experiments 1–4 performed poorly on test data when compared to experiments 5–8. It is unclear why this happened. More detailed error analysis is included in the subsection A.3.

From our experiments on 3 benchmark datasets, we observe that under the same network architecture and training settings, SplineCNN with original graclus-based pooling learns from WaveMesh

| # | Superpixel | #Nodes | Config | Pooling | Train acc | Test acc | Precision |
|---|---|---|---|---|---|---|---|
| 1 | WaveMesh | 197±82 | 1 | Graclus | 52.63±0.36 | 43.36±0.72 | 47.66±0.48 |
| 2 | WaveMesh | 197±82 | 1 | WavePool | 55.04±0.21 | **52.58±0.21** | **52.22±0.21** |
| 3 | WaveMesh | 197±82 | 2 | Graclus | 60.28±0.18 | 50.42±0.27 | 54.77±0.24 |
| 4 | WaveMesh | 197±82 | 2 | WavePool | 70.25±0.30 | **56.89±0.31** | **57.03±0.31** |
| 5 | SLIC | 215±15 | 1 | Graclus | 50.96±0.51 | 45.87±0.28 | 47.59±0.41 |
| 6 | SLIC | 215±15 | 2 | Graclus | 59.09±0.20 | **50.69±0.45** | **53.89±0.17** |
| 7 | SLIC (RingGNN) | 85–150 | 105165 | – | 19.56±16.40 | 19.30±16.12 | – |
| 8 | SLIC (MoNet) | 85–150 | 104229 | – | 65.92±2.52 | 54.66±0.52 | – |
| 9 | SLIC (Gated GCN) | 85–150 | 104357 | – | 94.55±1.02 | 67.31±0.31 | – |

Table 3: Results on CIFAR-10. Experiments 7–9 report min and max number of nodes in column 3, and the number of parameters in the model in column 4 (Dwivedi et al., 2020).

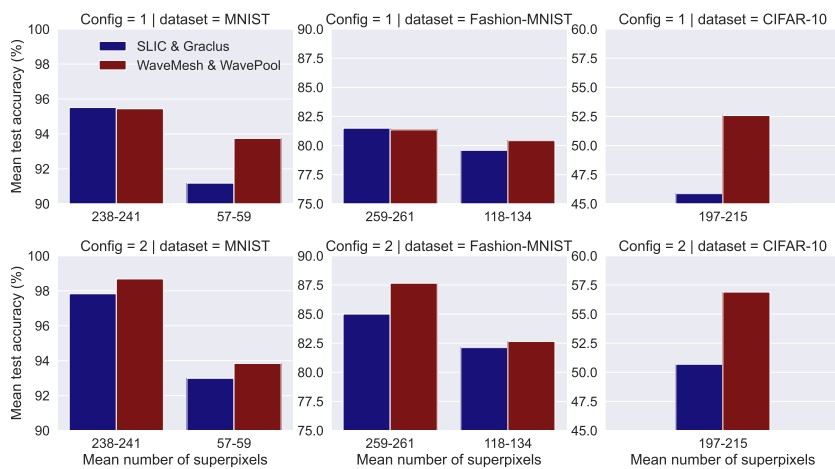

Figure 7: Mean test accuracy versus mean number of superpixels for all three datasets for both network configuration. The plot compares the accuracy of WaveMesh and WavePool combination with SLIC and Graclus combination.

superpixels on-par with SLIC superpixels. Additionally, under the same settings, we observe that the best performance is achieved when replacing graclus-based pooling with WavePool while using WaveMesh superpixels. This is shown in Figure 7 for all three datasets. We believe this increase in performance is because WavePool accounts for spatial heterogeneity while aggregating nodes. Overall, we conclude that WaveMesh is a reasonable sample-specific multiscale superpixeling method.

## 6 CONCLUSION

Over the last 5 years powerful GNNs have been developed for a variety of tasks on graph structured data. Nonetheless, for image graph classification tasks, GNN studies have been restricted to graphs that model a regular grid or similar-sized SLIC superpixel representations. Looking at images through the lens of GNNs enables rethinking the process of downsampling, and offers new possibilities for image representations that explore the landscape between the regular grid and similar-sized superpixel representations. Towards this goal, we introduced WaveMesh, a superpixeling algorithm that computes spatially heterogeneous superpixels of varying sizes within an image. We also proposed WavePool, a new pooling scheme tailored to WaveMesh superpixels. We investigate the performance of both methods across three benchmark datasets. Our experiments comparing WaveMesh superpixels with SLIC superpixels and WavePool with graclus-based pooling demonstrated promising results. Multiscale spatially heterogeneous superpixels warrant further attention. As a future direction, we encourage researchers to benchmark GNN models on WaveMesh superpixels and explore architectures custom to WaveMesh superpixels. In particular, we envision greater interest in this direction of research from the medical machine learning community where high resolution images are ubiquitous (*see subsection A.4*).

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

# A    APPENDIX

## A.1    DATASETS

1. MNIST: 28×28 grayscale images, 60k train and 10k test, 10 categories.
2. Fashion-MNIST: 28×28 grayscale images, 60k train and 10k test, 10 categories.
3. CIFAR-10: 32×32 color images, 50k train and 10k test, 10 categories.

SLIC superpixels were generated using the scikit-image library by setting the compactness parameter to 0.25 (van der Walt et al., 2014). A small value of compactness parameter was chosen to ensure that superpixels shapes are not all square. While computing WaveMesh superpixels MNIST and Fashion-MNIST images are padded with zeros to make them 32×32. See Figure 8 for examples of region adjacency graphs generated from CIFAR-10 images using WaveMesh superpixels.

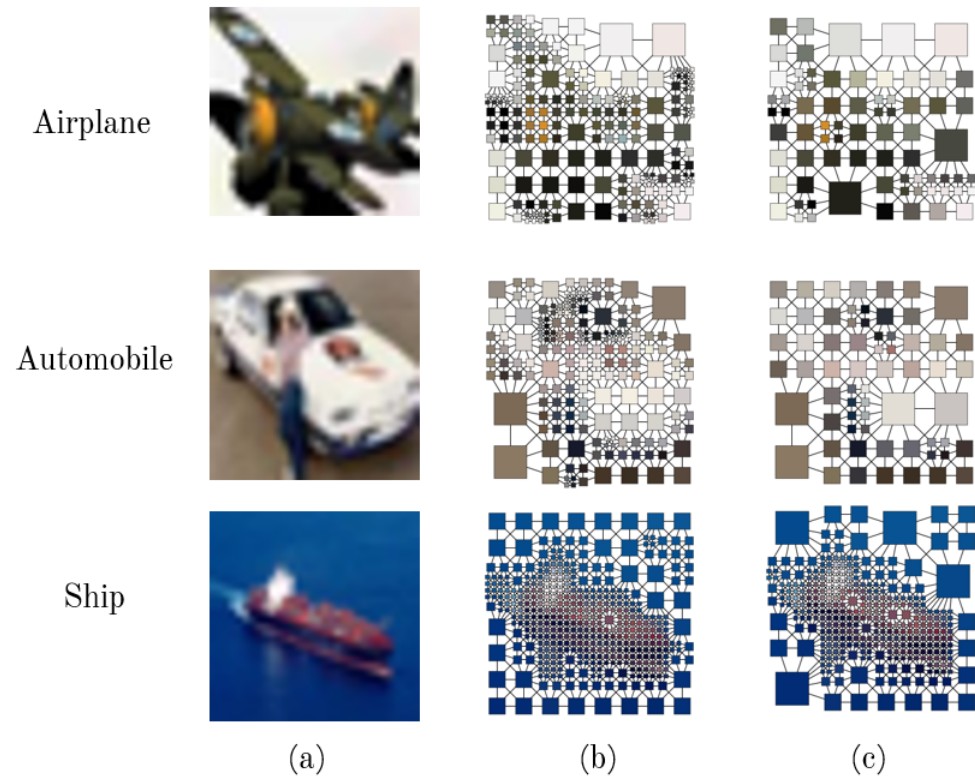

Figure 8: WaveMesh superpixels represented by region adjacency graphs (RAG). (a) Images from the CIFAR-10 dataset. (b) RAGs representing WaveMesh superpixels obtained using the theoretical threshold $T$ for images shown in (a). (c) RAGs representing WaveMesh superpixels obtained using a threshold equal to $2T$. Size of nodes in the graph are proportional to the corresponding superpixel size.

## A.2 ADDITIONAL NOTES ON STEP 2: FILTERING OF THE IMAGE IN WAVELET SPACE

Note that the energy-based, spatially local filter outlined in this study is fundamentally different from Fourier-based spectral filtering, in that the latter is a scale-sharp filter that acts globally in pixel space and does not allow the discrimination of localized, energetic structures. Additionally, unlike Fourier-based filters, the present method does not require periodicity.

## A.3 ERROR ANALYSIS

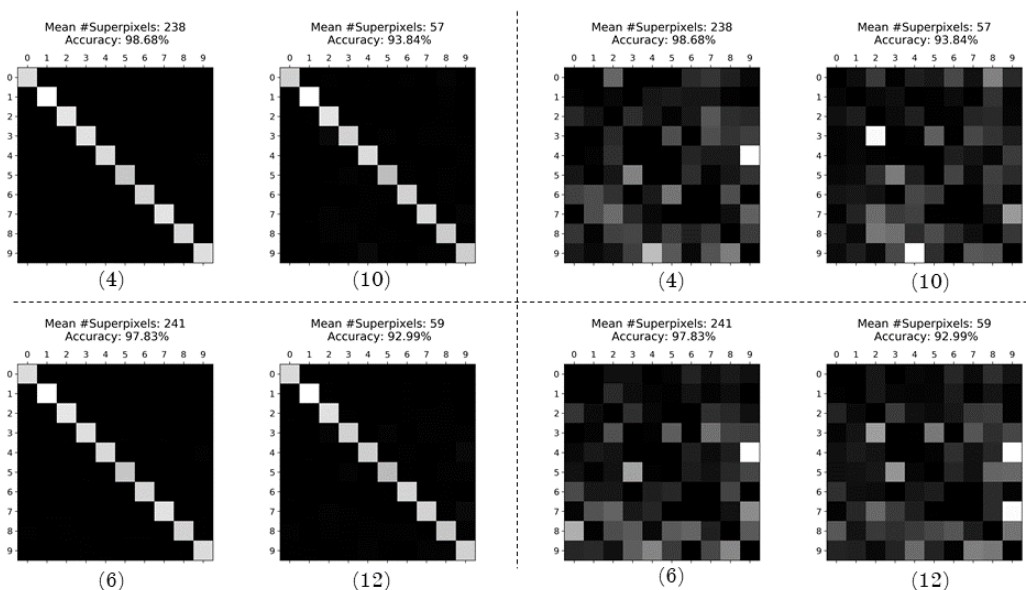

Figure 9: MNIST error analysis: The top row corresponds to experiments on WaveMesh superpixels with WavePool in config 2. The bottom row corresponds to experiments on SLIC superpixels with Graclus pooling in config 2. Left quadrant shows the confusion matrix and the right quadrant shows the error rate matrix averaged over all runs of an experiment. Experiment numbers from Table 1 are indicated below each matrix.

**MNIST**. In Figure 9, the confusion matrix looks good for all the experiments as most images are on the main diagonal. The actual class is along the rows and predicted class is along the columns. In Figure 9, the error rate matrix for each experiment in the right quadrant is obtained by dividing each value in the confusion matrix by the number of images in the corresponding class, and by filling the main diagonal with zeros.

- **SLIC with Graclus**. Comparing the error rate matrices of experiments 6 and 12, in both cases many digit 4 images are being misclassified as 9, and as the number of superpixel reduces, many digit 7 images are also being misclassified as 9.
- **WaveMesh with WavePool**. Comparing the error rate matrices of experiments 6 and 12In experiment 4 many digit 4 images are wrongly classified as 9. However, in experiment 10, with the number of superpixels equal to one-fifth of experiment 4, many 9s are being misclassified as 4 and many 3s are being misclassified as 2.

**FASHION-MNIST**. In Figure 10,

- from the confusion matrices, we can conclude that the model performs best in classifying images from the classes trouser, sandal, sneaker, bag, ankleboot. This is true both with WaveMesh and SLIC superpixels.
- from the error rate matrices we can conclude that shirt is getting misclassified the most. Also, the columns for classes shirt, coat and pullover are quite bright, indicating that many images are getting misclassified into these classes.

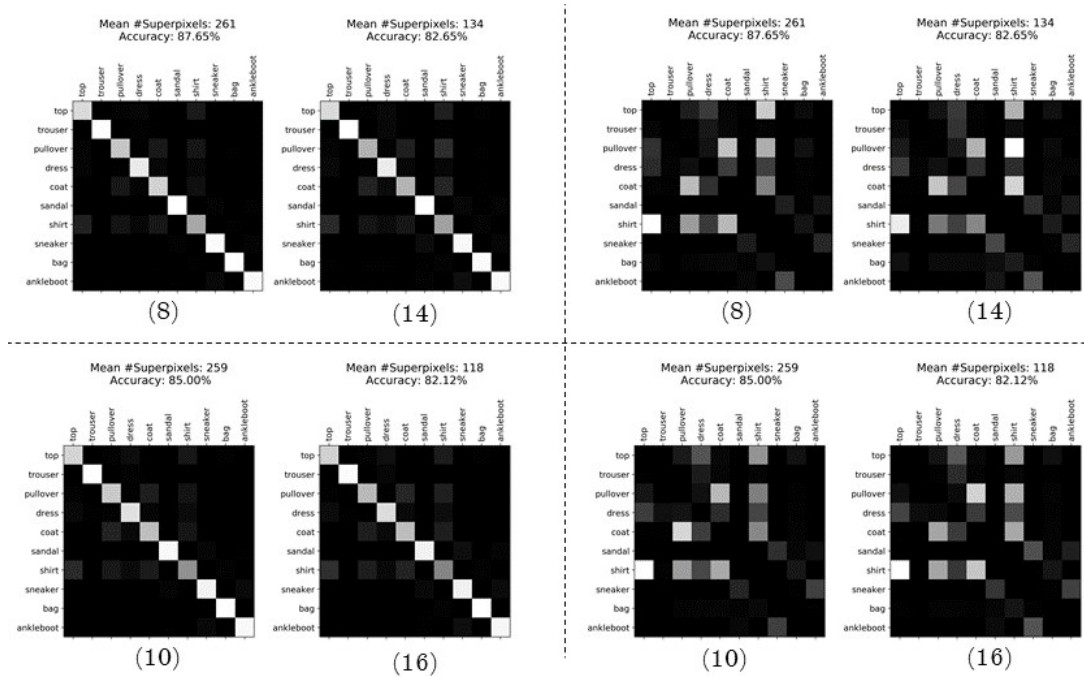

Figure 10: FashionMNIST error analysis: The top row corresponds to experiments on WaveMesh superpixels with WavePool in config 2. The bottom row corresponds to experiments on SLIC superpixels with Graclus pooling in config 2. Left quadrant shows the confusion matrix and the right quadrant shows the error rate matrix averaged over all runs of an experiment. Experiment numbers from Table 2 are indicated below each matrix.

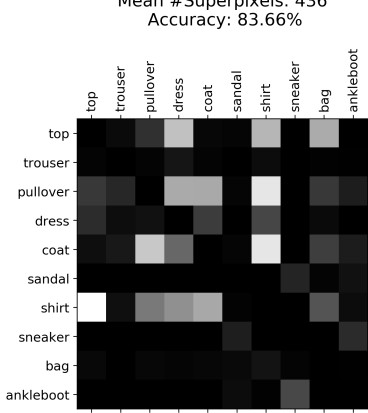

Figure 11: Error rate matrix for experiment 4 from Table 2. Comparing this matrix with that of experiment 8 from Figure 10, we observe that many more images of pullover and coat are getting misclassified as shirt in this experiment when compared to experiment 8.

Overall, from the error analysis for MNIST and Fashion-MNIST, we observe that misclassification patterns are not very different for WaveMesh and SLIC superpixels.

## A.4    WaveMesh Applied to Medical Images

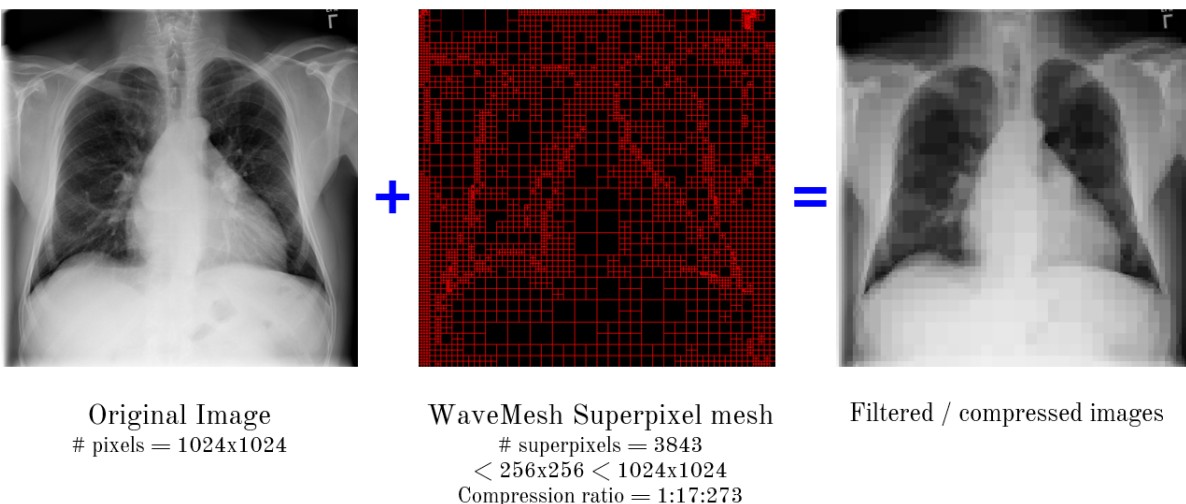

Original Image
# pixels = 1024x1024

WaveMesh Superpixel mesh
# superpixels = 3843
< 256x256 < 1024x1024
Compression ratio = 1:17:273

Filtered / compressed images

Figure 12: Left: Chest X-ray image of size 1024×1024. Center: WaveMesh superpixel mesh. Right: Wavelet filtered chest X-ray image.

The chest X-ray image shown in Figure 12 is from the NIH chest X-ray dataset (Wang et al., 2017). The image has 1024×1024 pixels. These X-ray images are typically downsampled to 256×256 before using them for training a CNN model. In the center in Figure 12, we show the WaveMesh superpixel mesh obtained using our algorithm. It has 3843 multiscale superpixels. We note that $3843 < 256^2 < 1024^2$. Infact, the compression ratio is $1 : 17 : 273$.

