# OpenReview forum: "Learning from multiscale wavelet superpixels using GNN with spatially heterogeneous pooling"
_ICLR.cc/2021/Conference — Reject_

### Official Review · AnonReviewer2 · 2020-10-23
**This paper proposes a multiscale superpixel algorithm that tries to circumvent the limitation of SLIC. It is novel but the experiments are not so convincing.**

**Rating:** 5
**Confidence:** 4

**Review:**

Quality: The motivation of this paper is great, and the proposed WaveMesh is interesting. In general, it is a high-quality work. It will be better to supplement more experiments.

Clarity: The expression is clear, and especially, the figures are very exquisite, and helpful for understanding.

Originality: The proposed image-specific superpixeling algorithm WaveMesh is novel and try to circumvent the limitation of SLIC. However, the insufficient experiments affect its persuasiveness.

Significance: The WaveMesh could filter the unimportant information and focus on the important information which is a significant research. It may be applied to more image-based task, such as image segmentation. Consequently, I think this work has a certain significance, but it needs more experiments to evaluate.

Pros:
1. It is a novel idea that non-uniformly downsamples the images and gets a multiscale superpixel representation.
2. The figures are concise and precise, and they provide an intuitive understanding about the WaveMesh and WavePool.

Cons:
1. The experiment is not sufficient. Take MNIST as example, this work only compares the WaveMesh and SLIC when the number of nodes is ~57 which is not convincing enough to conclude that the WaveMesh is effective. What is the acc about SLIC when the number of nodes is ~238. Besides, it seems that SLIC+SplineCNN achieves the best acc when the number of nodes is small. The experiments on Fashion-MNIST and CIFAR-10 have the same problem.
2. WaveMesh is a superpixeling algorithm that is not coupled with classification algorithm, but all the experiments are performed on SplineCNN. What is the performance of WaveMesh with other GNN algorithms?
3. How to evaluate the effectiveness of the image-dependent threshold T?

---

> ### Author Response · Authors · 2020-11-22
> **Response and pointers to modifications in revised manuscript**
>
> Thank you for the feedback, which has been helpful in improving the manuscript. Please refer to our responses below.
>
> > **1. It is a novel idea that non-uniformly downsamples the images and gets a multiscale superpixel representation.\
> > 2. The figures are concise and precise, and they provide an intuitive understanding about the WaveMesh and WavePool.**
>
> Thank you for the positive comments.
>
> > **1. The experiment is not sufficient. Take MNIST as example, this work only compares the WaveMesh and SLIC when the number of nodes is ~57 which is not convincing enough to conclude that the WaveMesh is effective. What is the acc about SLIC when the number of nodes is ~238. Besides, it seems that SLIC+SplineCNN achieves the best acc when the number of nodes is small. The experiments on Fashion-MNIST and CIFAR-10 have the same problem.**
>
>    * We have added results from 2 new experiments on MNIST with ~238 SLIC superpixels (see #5-6 in Table 1). The results from these confirm our previous conclusions.
>    * We have also added results from 6 new experiments on FashionMNIST (see #11-16 in Table 2). The results from all these confirm that SplineCNN is able to learn from WaveMesh superpixels on par or better than with SLIC superpixels.
>    * The best results are consistently obtained with (WaveMesh + WavePool) and larger number of nodes, except for an outlier set of experiments with Fashion-MNIST (see Tables 1-3 and relevant comments in the results section). To facilitate extraction of this conclusion, we have added a new figure (see Figure 7).
>    * We were unable to generate ~436 superpixels for FashionMNIST using the SLIC implementation in the skimage library. This is because of how the SLIC algorithm works for the image size under consideration. Our attempts resulted in at most ~250 superpixels. Hence, we couldn't perform SLIC experiments with ~436 superpixels for this dataset.}
>
> >  **2. WaveMesh is a superpixeling algorithm that is not coupled with classification algorithm, but all the experiments are performed on SplineCNN. What is the performance of WaveMesh with other GNN algorithms?**
>
>    * We decided not to evaluate the performance of WaveMesh with other GNN algorithms at the moment, as the goal of the present work is to study the feasibility of learning from WaveMesh -- a new multiscale superpixel representation -- instead of classical SLIC representations for image graph classification tasks (as opposed to a model development study whose goal would have been to reach/claim state of the art performance). We clarify these goals in the introduction section of the manuscript.
>    * We chose SplineCNN as baseline GNN for multiple reasons: i) it is a state-of-the-art GNN model for image graph classification, ii) Fey et al. (2018) reported that edge detecting patterns are well learned by the kernels in SplineCNN when trained on image superpixels, and iii) SplineCNN convolution operations possess the interesting property of mimicking classical CNN convolutions when the number of superpixels is the same as the number of pixels in the original image (which combines nicely with WavePool that naturally possesses the property of collapsing to classical CNN pooling operations for uniform inputs). We believe these reasons make SplineCNN a good candidate for this feasibility study.
>    * Our study systematically shows that it is possible to successfully learn from a multiscale representation on-par or better than from a single-scale representation. This now opens the door for benchmarking other models on this new representation, exploring different multiscale representations, and calls for more expressive GNNs.
>
> > **3. How to evaluate the effectiveness of the image-dependent threshold T?**
>
>    * The effectiveness of the image-dependent threshold T in generating image-dependent numbers of superpixels is shown in Figure 1b.
>    * The focus of this study is on learning performance, which we report in Tables 1-3 for varying mean number of superpixels (which surprisingly is rarely varied in similar studies), obtained by varying the threshold around the theoretical value described in Section 2.
>    * The effect of the image-dependent threshold T on the structural similarity of the compressed/superpixeled image vs that of the original image could also be measured using the SSIM metric. During our preliminary investigation, we systematically observed that the SSIM is higher with WaveMesh than with SLIC at a similar mean-number of superpixels. For example, the dataset average SSIM is 100% with WaveMesh vs 93.9% with SLIC for MNIST with ~238 superpixels, and 85.8% with WaveMesh vs 82.3% with SLIC for CIFAR-10 with ~200 superpixels. We chose not to focus on this class of metrics in this study to remain focused on the quality of the different superpixel representation for learning, which depends not only on the first step of the superpixel construction but also on subsequent ones such as building of the region adjacency graph.

---

### Official Review · AnonReviewer4 · 2020-10-28
**Learning from multiscale wavelet superpixels using GNN with spatially heterogeneous pooling**

**Rating:** 2
**Confidence:** 4

**Review:**

This paper present WaveMesh, a wavelet-based superpixeling algorithm and WavePool, a spatially heterogeneous pooling scheme.

Strengths:

"A wavelet transform image coding technique with a quadtree structure" (Banham and Sullivan 1992) and "Geometric methods for wavelet-based image compression" (Wakin et al. 2003) indicate that a wavelet-coefficient based quadtree decomposition of images is an efficient image-compression technique. This suggests that the goal of this paper is reasonable and the method can likely be motivated.

Weakness:

The authors provide little theoretical justification for the use of wavelets as they are used in this paper.

Moreover, the pooling method in this paper appears to be the same as that in "Image Segmentation Based on Multiscale Fast Spectral Clustering". In addition, the pooling method seems to be applicable to any quadtree-based method. A superpixel quadtree can be defined in many ways, and quadtrees are the basis of other superpixel methods like the "A Hierarchical Data Structure for Picture Processing" (Tanimoto and Pavlidis 1975), analyzed in "Evaluation Framework of Superpixel Methods with a Global Regularity Measure."

There are two primary weaknesses: First, SLIC is a poor baseline and many other superpixel methods have been developed since 2010 - "Superpixel Segmentation using Linear Spectral Clustering" or "Robust superpixels using color and contour features along linear path" or even the recent "Simple and fast image superpixels generation with color and boundary probability".
Second, almost all of the gains of this method appear to come from the pooling method: In Tables 2 and 3, WaveMesh alone consistently underperforms SLIC.

Third, these results appear to depend on the classification model. While the variance is low, alternative hyperparameters for the SplineCNN (or a different model) may have resulted in different performance. Given that this paper did not use any of the existing superpixel evaluation metrics, it is difficult to compare it to prior work. While of course the two papers cannot be compared directly as they are both currently under ICLR review, "Probabilistic Numeric Convolutional Neural Networks," has similar performance on the 75 Superpixel MNIST dataset, using a third as many superpixels.

Due to these weaknesses, this paper has neither the experimental analysis nor the theoretical grounding to be appropriate for ICLR at this time.

---

> ### Author Response · Authors · 2020-11-22
> **Response 1/2 and pointers to modifications in revised manuscript**
>
> ## Part 1/2:
>
> Thank you for the feedback, which has been helpful in improving the manuscript. We believe there were some misunderstandings that we clarified below and in relevant sections of the manuscript.
>
> > **"A wavelet transform image coding technique with a quadtree structure" (Banham and Sullivan 1992) and ``Geometric methods for wavelet-based image compression" (Wakin et al. 2003) indicate that a wavelet-coefficient based quadtree decomposition of images is an efficient image-compression technique. This suggests that the goal of this paper is reasonable and the method can likely be motivated.**
>
> Thank you for the positive comment.
>
> > **The authors provide little theoretical justification for the use of wavelets as they are used in this paper.**
> - The theoretical grounding of the method is described throughout Section 3, which also contains references to other works that have used wavelets in a similar way. We have incorporated an additional sentence with references to the work by Banham & Sullivan (1992) and Wakin et al. (2003) since their conclusions justify some of the choices made in the method's part of this study.
> - We recognize that the way we use wavelets in this paper is different from the classical ways in which wavelet theory is used in the machine learning community. Our use of wavelets follows how they have been successfully used in other scientific disciplines, e.g. in the physical sciences. Because learning from the pixel space representation of images remains the optimal method to date, we believe cross-fertilizing from other uses of wavelet theory consistent with this representation to be promising, and one of the ways by which our work distinguishes itself from other works in the ML community using the wavelet transform.
>
> > **Moreover, the pooling method in this paper appears to be the same as that in "Image Segmentation Based on Multiscale Fast Spectral Clustering".**
>
> - After careful review of the manuscript by Zhang et al. (2018), we are confident that our proposed WavePool method is different and remains novel (both in its application and implementation). Nonetheless, we have added a clarifying sentence in the revised submission and cited the manuscript (see page 6, sec 4 both paragraphs). See below for more details:
> - In their work, Zhang et al. (2018) describe a quadtree-based *segmentation* algorithm that uses a superpixel *similarity matrix* to repeatedly merge child nodes from the finest level *all the way* to the coarsest level using a distinct *fast spectral clustering* method. When the coarsest level is reached, the segmented image is obtained whose *resolution is the same as that of the original image*.
> - In contrast, in our work, once we compute WaveMesh superpixels for an image, we build an embedded region adjacency graph (RAG) to represent the image. This RAG is not a tree (for example see the graphs in Fig 2, and in the lower panel of Fig 4), and should not be confused with the wavelet coefficient quadtree. Additionally, WavePool is implemented in our work as a layer in a GNN to perform a single-step rule-based graph contraction. When merging superpixels, WavePool also aggregates node features with a choice of pooling function (mean or max typically).
>
> > **In addition, the pooling method seems to be applicable to any quadtree-based method. A superpixel quadtree can be defined in many ways, and quadtrees are the basis of other superpixel methods like the "A Hierarchical Data Structure for Picture Processing" (Tanimoto and Pavlidis 1975), analyzed in "Evaluation Framework of Superpixel Methods with a Global Regularity Measure."**
>
> Thank you for pointing this out. We agree that WavePool is not restricted to WaveMesh superpixels but is broadly applicable to any superpixel representation that can be directly (as in Zhang et al. 2018) or indirectly (as in this work via surrogate wavelet coefficient quadtree) be represented using a quadtree. We believe this generalizability is an advantage of the method. We have added clarifying sentences and the reference (see page 6, last paragraph).

---

> > ### Author Response · Authors · 2020-11-22
> > **Response 2/2 and pointers to modifications in revised manuscript**
> >
> > ## Part 2/2
> >
> > > **There are two primary weaknesses: First, SLIC is a poor baseline and many other superpixel methods have been developed since 2010 - "Superpixel Segmentation using Linear Spectral Clustering" or "Robust superpixels using color and contour features along linear path" or even the recent ``Simple and fast image superpixels generation with color and boundary probability". Given that this paper did not use any of the existing superpixel evaluation metrics, it is difficult to compare it to prior work.**
> >
> > - We compare our results with SLIC because we wanted to compare with existing literature from the graph learning community, since our primary goal is to establish the feasibility of learning from embedded graphs that represent multiscale spatially heterogeneous superpixels. GNN papers typically benchmark models for image graph classification on embedded graphs formed from similar-sized SLIC superpixels. Nonetheless, we have cited the aforementioned papers in the revised manuscript (see page 2, superpixeling paragraph).
> >
> > > **Second, almost all of the gains of this method appear to come from the pooling method: In Tables 2 and 3, WaveMesh alone consistently underperforms SLIC.**
> >
> > - We agree with your second observation. However, we believe that since WavePool is not applicable to SLIC superpixels, the consistent improvement observed between (SLIC + Graclus) and (WaveMesh + WavePool) cannot be attributed to WavePool alone. We added an additional figure (see Figure 7) to clarify this conclusion.
> >
> >
> > - Furthermore, the GNN used in this study was originally designed and tested with SLIC superpixels. Topics of future research include investigating novel graph convolutional layers tailored to WaveMesh or other multiscale superpixels.
> >
> > > **Third, these results appear to depend on the classification model. While the variance is low, alternative hyperparameters for the SplineCNN (or a different model) may have resulted in different performance. While of course the two papers cannot be compared directly as they are both currently under ICLR review, ``Probabilistic Numeric Convolutional Neural Networks," has similar performance on the 75 Superpixel MNIST dataset, using a third as many superpixels.**
> >
> > - Like in (Dwivedi et al., 2020), our goal is not to find the optimal set of hyperparameters to achieve state-of-the-art results. To be rigorous and fair, it would require an exhaustive hyperparameter search for all the experiments reported in the paper, which in turn would be computationally expensive. In fact, in the experiments, we use the same network architecture and hyperparameters as those used by the authors of the method we are comparing against (e.g. as in Fey et al., 2018). Therefore we believe the results for our methods should be considered conservative baselines. Note this is in contrast to what appears to be the objective of ``Probabilistic Numeric Convolutional Neural Networks" that targets a carefully tuned state-of-the-art performance (using classical SLIC superpixels as input).
> >
> >
> > - For similar reasons, we decided not to evaluate the performance of WaveMesh with other GNN algorithms at the moment, as the goal of the present work is to study the feasibility of learning from WaveMesh -- a new multiscale superpixel representation -- instead of classical SLIC representations for image graph classification tasks (as opposed to a model development study whose goal would have been to reach/claim state of the art performance). We clarify these goals in the introduction section of the manuscript.
> >
> >
> > - We chose SplineCNN as baseline GNN for multiple reasons:
> >    1. It is a state-of-the-art GNN model for image graph classification.
> >    2. Fey et al. (2018) reported that edge detecting patterns are well learned by the kernels in SplineCNN when trained on image superpixels.
> >    3. SplineCNN convolution operations possess the interesting property of mimicking classical CNN convolutions when the number of superpixels is the same as the number of pixels in the original image (which combines nicely with WavePool that naturally possesses the property of collapsing to classical CNN pooling operations for uniform inputs).
> >
> > We believe these reasons make SplineCNN a good candidate for this feasibility study.

---

### Official Review · AnonReviewer3 · 2020-10-28
**introduces a wavelet-based superpixel algorithm**

**Rating:** 5
**Confidence:** 3

**Review:**

This paper introduces a wavelet-based superpixel algorithm and a spatially heterogeneous pooling. More specifically, they introduce an algorithm to compress images in the pixel domain and it leads non-uniformly distributed and multiscale superpixels. Furthermore, they introduced a spatially heterogeneous pooling method tailored to the superpixel algorithm. Finally, they demonstrate the effectiveness of their method on MNIST, Fashion-MNIST, and CIFAR-10 datasets.
pros:
1. The proposed method is simple and elegant. It is theoretical well-founded and easily implemented.
2. The paper provides good initial results, showing that to some degree their method is generalizable.
cons:
1. In step 2 of wavemesh, the authors need to include the computational cost.
2. In Table1, the results of the proposed method are not significant. More experiments are needed to demonstrate the gain of the proposed method.
3. The authors should try other evaluation measurements, only acc is not enough.
4. I believe more generalization and deeper analysis would be beneficial.

---

> ### Author Response · Authors · 2020-11-22
> **Response and pointers to modifications in revised manuscript**
>
> Thank you for the feedback, which has been helpful in improving the manuscript.
>
> > **1. The proposed method is simple and elegant. It is theoretical well-founded and easily implemented.\
> > 2. The paper provides good initial results, showing that to some degree their method is generalizable.**
>
> Thank you for the positive comments.
>
> > **3. In step 2 of wavemesh, the authors need to include the computational cost.**
>
> Thank you for pointing this out. We have included the computational cost (see page 5, paragraph 1).
>
> > **4. In Table 1, the results of the proposed method are not significant. More experiments are needed to demonstrate the gain of the proposed method.**
>
> We have added results from 8 new experiments (see #5-6 in Table 1 and #11-16 in Table 2) and a new figure to facilitate comparison of the methods (see Figure 7). All new results confirm our previous conclusions.  The best results are consistently obtained with (WaveMesh + WavePool) . To the best of our knowledge, "learning from multiscale spatially heterogeneous superpixels using GNNs" hasn't been studied in this way before. Our study shows that it is both feasible and promising, which opens the door for benchmarking other models on this new representation, exploring different multiscale representations, and calls for more expressive GNNs.
>
> > **5. The authors should try other evaluation measurements, only acc is not enough.\
> > 6. I believe more generalization and deeper analysis would be beneficial.**
>
> We have added precision for all the experiments in Tables 1-3. We didn't include recall since averaged one-versus-all recall and accuracy are equal for balanced datasets. The values of all reported metrics in Tables 1-3 consistently support our conclusions (we also refer you to the new experiments mentioned above). In addition, we also added a section on Error Analysis in the appendix (see Page 14 Section A.3).

---

### Official Review · AnonReviewer1 · 2020-10-30
**Good paper**

**Rating:** 7
**Confidence:** 4

**Review:**

The paper introduces a new approach to leveraging graph neural networks for image tasks. While prior work has been based on constructing graphs using a super-pixel map using methods like SLIC, that generate super-pixels of all roughly the same size, the proposed method generates a scale-adaptive partition. It also introduces a new pooling operator on this graph structure, and demonstrates that, together, these lead to improved performance when used with GNNs.

- Overall, the method is novel and innovative, and well executed. The experimental evaluation is reasonably compelling (even if on relatively small datasets).

- This paper adds to the body of literature exploring the use of GNNs as an alternative to CNNs for image tasks. The fact that past attempts have relied on a graph structure built at a specific scale likely handicap them when competing with CNNs, which are much better able to retain notion of space and scale across successive convolution and pooling operations.

- The idea of converting a tagged/thresholded wavelet coefficients into a quad-tree structure is interesting, and indeed natural in retrospect.

- I wonder though whether alternative approaches to building a multi-scale superpixel partition would perhaps not perform better. This isn't a negative for this paper (indeed, it could spur such research in the future), but perhaps something that's based on hierarchical clustering (on top of SLIC) might work better. One drawback of the wavelet approach is that the boundaries of the superpixels have to be regular and axis and aligned with the downsampling phase (have authors considered variants that are based on overcomplete wavelet transforms?).

- There is an interesting effect in Fashion-MNIST where a larger number of superpixels leads to lower accuracy. This doesn't seem to have been discussed in the text.

- Minor comment: For Fig 3, it would be good to have a brief legend or description in the caption itself. The caption currently points to Sec 3.3 (which is a little inconvenient when parsing the figure).

---

> ### Author Response · Authors · 2020-11-22
> **Response and pointers to modifications in revised manuscript**
>
> Thank you for the feedback, which has been helpful in improving the manuscript.
>
> > **Overall, the method is novel and innovative, and well executed. The experimental evaluation is reasonably compelling (even if on relatively small datasets).**
>
> > **This paper adds to the body of literature exploring the use of GNNs as an alternative to CNNs for image tasks. The fact that past attempts have relied on a graph structure built at a specific scale likely handicap them when competing with CNNs, which are much better able to retain notion of space and scale across successive convolution and pooling operations.**
>
> > **The idea of converting a tagged/thresholded wavelet coefficients into a quad-tree structure is interesting, and indeed natural in retrospect.**
>
> Thank you for the positive comments. To further strengthen the experimental evaluation, we have added results from 8 new experiments (see #5-6 in Table 1 and #11-16 in Table 2), and we have also reported additional evaluation metrics (see Tables 1-2, Figure 7, and Appendix A.3). All the new results confirm our previous conclusions. Please refer to our point-by-point responses below.
>
> > **I wonder though whether alternative approaches to building a multi-scale superpixel partition would perhaps not perform better. This isn't a negative for this paper (indeed, it could spur such research in the future), but perhaps something that's based on hierarchical clustering (on top of SLIC) might work better.**
>
> The WaveMesh method is certainly not the only possible way to build a multiscale superpixel partition, and we expect future works to attempt similar goals from SLIC alone. For example, we cited the work of Knyazev et al. (2019) in our original submission to reflect an example of such other exploratory work. In their work, Knyazev et al. (2019) construct multi-graphs that represent uniform-sized SLIC superpixels computed at different user-defined scales. In contrast, WaveMesh constructs a single graph with spatially heterogeneous superpixels of various sizes determined dynamically. To rigorously design an optimal multiscale superpixel learning representation remains an open research question, and we hope our study will contribute to instigating such research.
>
> > **One drawback of the wavelet approach is that the boundaries of the superpixels have to be regular and axis-aligned with the downsampling phase (have authors considered variants that are based on overcomplete wavelet transforms?).**
>
> We agree that regular and axis-aligned superpixels is a property of WaveMesh superpixels. We have added a sentence to clarify this aspect (see page 5, paragraph 1). This partly results from relying on the fully discrete wavelet transform. We have not considered a variant based on overcomplete wavelet transforms, which are generally considered unnecessarily computationally expensive due to the inherent redundancy in continuous wavelet coefficient values. However, more advanced methods that circumvent this potential drawback certainly warrant further investigation. Examples include other types of multiscale geometric transforms. We have added an additional paragraph to highlight some of these alternatives (see page 4, paragraph 3).
>
> > **There is an interesting effect in Fashion-MNIST where a larger number of superpixels leads to lower accuracy. This doesn't seem to have been discussed in the text.**
>
> We ran additional experiments on Fashion-MNIST (see #11-14 in Table 2) and confirmed that this is not a trend, which suggests this is an outlier set of experiments. We have added relevant comments (see page 8, paragraph on Fashion-MNIST), as well as a section on error analysis in the appendix (see page 14, section A.3).
>
> > **Minor comment: For Fig 3, it would be good to have a brief legend or description in the caption itself. The caption currently points to Sec 3.3 (which is a little inconvenient when parsing the figure).**
>
> Thank you for pointing this out. We corrected this by adding a more comprehensive description in the figure caption.

---

### Decision · Program_Chairs · 2021-01-07
**Final Decision**

**Decision:**

Reject

**Comment:**

This paper received mixed reviews.  One reviewer is positive, while the remaining three reviewers are either negative or feel that the paper is below the threshold for acceptance.

The ideas presented in the paper are interesting and novel - this was acknowledged by three of the reviewers, even those who did not recommend acceptance.  The AC also recognizes the technical novelty presented.  However, as all the reviewers pointed out to varying degrees, the experimentation is problematic and the AC in agreement with this.  In particular, the heavy focus on improvement on top of SLIC makes it the applicability of the proposed approach highly limited and also not so convincing.

Recommendation for the paper is to reject and resubmit with improved experimentation.